# OpenReview forum: "SafeEditor: Unified MLLM for Efficient Post-hoc T2I Safety Editing"
_ICLR.cc/2026/Conference — ICLR 2026 Conference Withdrawn Submission_

### Official Review · Reviewer_nZA5 · 2025-10-30

**Soundness:** 2
**Presentation:** 3
**Contribution:** 1
**Rating:** 2
**Confidence:** 4

**Summary:**

The paper introduces SafeEditor, an MLLM-based framework for safe image generation. The key idea is to prevent the production of unsafe images in text-to-image pipelines by using a custom-trained MLLM that performs multi-round editing until the result is safe. SafeEditor is trained on a dedicated dataset built for this purpose, consisting of interleaved images and textual explanations in a multi-round format. The paper compares the method with existing approaches based on prompt filtering, prompt modification, and output filtering. It also includes ablation studies on the safety–utility trade-off and an analysis of the effect of multiple editing rounds.

**Strengths:**

- The paper clearly explains the problem setup and is well written and easy to follow.
- I appreciate the effort invested in building the MR-SafeEdit benchmark, which could represent a valuable contribution to the community. Its construction progressively increases the safety of output images, making it potentially useful for advanced training strategies.

**Weaknesses:**

My main concern with the manuscript lies in the motivation and design of the proposed approach.

1. Safety methods for text-to-image generation can generally be divided into two categories: methods for hosted models and methods for open-source models. Hosted-model approaches assume that users have access only through an API and include strategies such as prompt filtering and image analysis. When users have direct access to the model, these techniques become ineffective, as they can be easily disabled. In such cases, alignment methods at the weight level are preferred, as correctly mentioned in Section 2. Based on its functioning, SafeEditor falls within the category of hosted-model methods. When dealing with hosted models, we assume that a centralized server runs the text-to-image system and serves users through inference requests. Each inference incurs a computational cost for the host. In the case of SafeEditor, the model may need to be executed multiple times (up to four rounds) before producing an acceptable image. Consequently, SafeEditor involves significantly higher inference costs compared to approaches like GuardT2I or Latent Guard. Despite this, the paper does not discuss the computational or economic implications of this design choice, which I consider a major limitation that should be explicitly addressed.
2. Related to the previous point, I find the motivation for addressing safety in text-to-image models through this approach unclear. If the model is hosted and a user attempts to generate an image containing banned or harmful content, the natural response would be to notify or restrict the user rather than repeatedly generate alternative images that consume significant GPU resources while still resembling the original request. It is worth noting that other approaches, such as Safe Latent Diffusion [1], achieve safety without introducing additional computational overhead.
3. The evaluation and comparison with baselines appear suboptimal. In Table 1, the paper reports only the refusal rate, which alone provides little insight into whether the method is functioning as intended. Disabling all safety mechanisms would trivially result in perfect metrics, yet this would be meaningless. It is necessary to actually generate the images and assess both their safety and their alignment with the user’s request. Moreover, input filtering methods rely on a threshold to classify prompts as safe or unsafe. Instead of reporting absolute values, it would be more informative to present performance curves as a function of this threshold. Finally, I believe that LlamaGuard2 is a missing baseline in the prompt-filtering comparison, as it is a widely used reference in this context.
4. Finally, unless I am missing something, the results do not appear particularly compelling. The absolute performance is quite similar to that of certain baselines, such as GuardT2I in Table 1 and SAFREE in Table 2, which achieve comparable outcomes with significantly lower computational requirements.

As a minor note, the related work section is rather limited and discusses only a small number of methods. I suggest expanding it to include approaches for both closed-source and open-source text-to-image models (for example, concept-forgetting techniques), and to dedicate a subsection to the use of MLLMs for harmful concept detection. This is only a suggestion, but any restructuring that provides a clearer and more complete overview of the state of the art would be beneficial.

[1] Safe Latent Diffusion, CVPR 2023

**Questions:**

1. How is the latency and computational cost of SafeEditor compared to baselines?
2. Why is generating intermediate images progressively useful rather than interrupting the generation when a malicious intent is detected?
3. How does the model perform compared to prompt/output filtering strategies in terms of image safety and quality?
4. How do baselines perform if we variate the classification threshold for safe/unsafe?
5. How do LLaMAGuard2 perform on the same task?
6. Is it possible to justify the low performance gap with baselines?

---

### Official Review · Reviewer_apfP · 2025-11-01

**Soundness:** 3
**Presentation:** 3
**Contribution:** 3
**Rating:** 4
**Confidence:** 3

**Summary:**

This paper investigates the safety alignment of Text-to-Image (T2I) models, focusing on the prevalent challenges of over-refusal and the trade-off between safety and utility. The authors propose a novel post-hoc editing framework named SafeEditor, at the core of which is a unified Multimodal Large Language Model (MLLM) capable of performing multi-round, iterative edits on unsafe images until they meet safety requirements. To train this model, the paper introduces MR-SafeEdit, a large-scale, multi-round, image-text interleaved safety editing dataset. Experimental results demonstrate that this approach surpasses existing methods in significantly reducing over-refusal rates and achieving a more favorable safety-utility balance.

**Strengths:**

1. The proposed "post-hoc safety editing" is a highly innovative and practical paradigm. It mimics the human cognitive process of identifying and refining unsafe content, directly addressing the "one-size-fits-all" and over-refusal issues of existing filtering or prompt modification methods, which is critical for improving user experience.

2. The construction of the MR-SafeEdit dataset is a major contribution of this work. This large-scale dataset, comprising 27,253 multi-round editing instances, provides the community with a valuable resource that can drive future research in more fine-grained and context-aware safety alignment.

3.  The paper's experimental design is exceptionally thorough. The authors conducted a comprehensive comparison of SafeEditor against two major classes of baselines (filter-based and prompt-modification methods) across multiple standard datasets. The evaluation metrics cover multiple dimensions, including over-refusal, safety, and utility (e.g., CLIP score, LPIPS score), robustly demonstrating the proposed method's effectiveness and superiority. Furthermore, extensive ablation studies offer deep insights into the contribution of each component of the model.

**Weaknesses:**

1. While the multi-round iterative editing paradigm is effective, it may introduce significant inference latency and computational overhead compared to single-pass filtering methods. The paper fails to provide an analysis of inference time or computational cost, which is crucial for the method's practical deployment. A discussion on the efficiency-performance trade-off is recommended.

2. The synthesis pipeline for the MR-SafeEdit dataset relies on GPT-4o. This dependency on a powerful, closed-source model raises concerns about the reproducibility of the dataset creation process and the potential for inheriting the "teacher model's" latent biases. Exploring the feasibility of using open-source models for similar dataset construction would enhance the work's value.

3. As the authors acknowledge, the definition of "safety" in this work is primarily confined to six common categories (e.g., sexual, violence, hate). The model's effectiveness on more complex, culturally sensitive, or subtle forms of harmful content (e.g., nuanced misinformation, politically sensitive topics) remains unverified.

4. The paper claims the method edits images with minimal utility loss, validated by metrics like the CLIP score. However, multi-round iterative editing still poses a risk of deviating from the user's original, subtle intent, especially on complex or artistic prompts. Relying solely on automated metrics may not fully capture these semantic nuances. Incorporating human evaluation to assess the preservation of user intent would make the conclusions more persuasive.

**Questions:**

Refer to weaknesses.

---

### Official Review · Reviewer_kZVF · 2025-11-03

**Soundness:** 2
**Presentation:** 2
**Contribution:** 2
**Rating:** 4
**Confidence:** 3

**Summary:**

This paper introduces SafeEditor, a novel framework for enhancing the safety of T2I models. The core idea is to move away from traditional filtering (which often leads to over-refusal) and pre-hoc prompt modification (which can compromise user intent) towards a post-hoc safety editing paradigm.

**Strengths:**

Pros:
1. They propose a multi-round, post-hoc editing process where an unsafe generated image is iteratively refined until it meets safety standards, rather than being outright rejected.
2. To train a model for this task, they constructed a large-scale, multi-round, image-text interleaved dataset.
3. The experiments are comprehensive and the results are compelling.

**Weaknesses:**

Cons:
1. The technical contribution is unclear. It seems this paper only proposes a pipeline and directly adopts the exiting editing methods. This paper does not propose any new editing methods for T2I safety.
2. The multi-round, iterative nature of SafeEditor (generate -> evaluate & edit -> potentially repeat) could introduce significant latency. For a real-time user-facing application, this could be a major bottleneck. The paper lacks any discussion or measurement of inference speed, which is a critical factor for practical deployment.
3. The paper shows that CLIP scores are largely preserved, but multiple rounds of editing could still cause the final image to drift from the user's original, nuanced intent. The framework seems optimized for clear safety violations, but its behavior on prompts with complex or subjective meanings (e.g., historical art depicting violence, subtle satire) is not explored. A discussion of these "semantic failure modes" would be beneficial.
4. The data generation pipeline uses a "refined prompt" to generate the next image in the sequence. However, the SafeEditor model itself seems to perform direct image-to-image editing mediated by textual thought. Is the refined prompt from the dataset used in any way during the training or inference of SafeEditor, or is it purely an artifact of the data creation process?
5. The paper rightly celebrates the extremely low false positive (over-refusal) rate. What is the corresponding false negative rate (i.e., the rate at which SafeEditor fails to edit an unsafe image and accepts it)? Understanding this trade-off is crucial for a complete picture of its safety performance.
6. Safety is often subjective. How does SafeEditor behave on borderline cases that might be considered unsafe by some but acceptable by others? Does the multi-round process allow for a "softening" of content rather than a complete removal, and how is that decision made?
7. The editing process stops when the model outputs "text only" (the "accept" decision). In practice, is there a hard limit on the number of rounds to prevent infinite loops or excessive latency on particularly difficult-to-edit images? The dataset statistics show up to four rounds; was this a natural limit or an imposed one?

**Questions:**

See Weaknesses

---

### Official Review · Reviewer_39Bq · 2025-11-06

**Soundness:** 1
**Presentation:** 1
**Contribution:** 1
**Rating:** 2
**Confidence:** 3

**Summary:**

This paper proposes a multi-round safety editing framework that mimics the human cognitive process of identifying and refining unsafe content. It includes a synthesis dataset MR-SafeEdit generated by GPT-4o and SD 3.5 and an MLLM SafeEditor trained on the MR-SafeEdit. The experiments show that SafeEditor could reduce the over-refusal rate.

**Strengths:**

- Important topic.
- Introduce human cognitive process into the identification and refining of unsafe content.

**Weaknesses:**

- This framework highly relies on GPT-4o-generated supervision signals without validating their reliability. All components of the training data MR-SAFEEDIT are generated by GPT-4o (thought, judgement, refined prompt) and SD 3.5 (re-generated images). That basically means the paper implicitly assumes that GPT-4o is accurate and consistent in both semantic understanding and safety classification of images. However, there is no quantitative or human-validated evidence provided to support this assumption, i.e., how do you prove the GPT-4o is reliable on this task? Moreover, the authors also annotate the safety labels of generated images using GPT-4o, which is also unreliable.

- Based on the training process, SafeEditor is basically mimicking GPT-4o's behavior. Given that GPT-4o already performs image safety reasoning and refined prompt generation, the same refinement workflow (GPT-4o -> refined prompt -> SD re-generation) can be directly executed at inference time. Therefore, it remains unclear why SafeEditor is needed in addition to GPT-4o rather than simply using GPT-4o + SD. If GPT-4o is reliable, SafeEditor is redundant; if GPT-4o is unreliable, SafeEditor inherits and amplifies those errors.

- This framework may silently substitute the outputs, which raises transparency and trust concerns. SafeEditor produces modified images that may differ semantically from original model outputs. The user receives the modified results without being informed that the content was altered. This implicit substitution of generated images may compromise user understanding of model behavior and introduce potential issues regarding transparency and trust between the user and the model provider.

- Several experimental design choices are confusing and require clarification. For example, SafeEditor is trained and deployed with SD 3.5 latents, but Section 5.2.2 appears to evaluate images generated by SD 1.4, whereas Section 5.2.1 uses SD 3.5. This inconsistency makes it difficult to interpret the utility evaluation and raises concerns regarding the fairness of the comparison. In addition, several evaluation metrics (e.g., high-level safety ratio, HP score, UIA score, CLIP score, and LPIPS score) require clearer definitions and justification for their relevance in this context.

- The additional computational cost introduced by SafeEditor is not reported. The method involves multiple stages, which likely incur significantly higher training and inference costs compared to baselines. However, the paper does not provide any measurement or discussion of computational overhead. Without such analysis, it is difficult to assess the practical feasibility and deployment value of the proposed framework, especially in comparison to simpler baselines.

- I recommend that the authors spend more time on improving the writing quality and fixing the typos.

**Questions:**

- How do you prove the GPT-4o is reliable on this task?
- Why is SafeEditor needed, since GPT-4o + SD can do the job?
- What is the extra computational cost?
- How do you address the transparency issue?

---

### Note · Authors · 2026-01-25

I have read and agree with the venue's withdrawal policy on behalf of myself and my co-authors.